# Quantification and Qualification of Floral Patterns of *Coffea arabica* L. in Colombia

**DOI:** 10.3390/plants12183332

**Published:** 2023-09-21

**Authors:** Carlos Andres Unigarro, Luis Carlos Imbachi, Aquiles Enrique Darghan, Claudia Patricia Flórez-Ramos

**Affiliations:** 1Plant Physiology, Centro Nacional de Investigaciones de Café, Manizales 170009, Colombia; 2Biometrics, Centro Nacional de Investigaciones de Café, Manizales 170009, Colombia; luis.imbachi@cafedecolombia.com; 3Facultad de Ciencias Agrarias, Universidad Nacional de Colombia, Bogotá 111321, Colombia; aqedarghanco@unal.edu.co; 4Plant Breeding, Centro Nacional de Investigaciones de Café, Manizales 170009, Colombia; claudia.florez@cafedecolombia.com

**Keywords:** coffee, flowering events, phenology, flowering synchrony, temporal variability

## Abstract

The phenological patterns of coffee flowering in Colombia have typically been studied in a descriptive way, with knowledge from an inferential perspective being scarce. The present study evaluated the effect of geographic location and accession on the floral patterns and phenological descriptors of *Coffea arabica* L. Fifteen accessions from the Colombian coffee collection (four tall and eleven short) were planted in the departments of Cesar, Caldas, Quindío and Cauca (Colombia). The number of flower buds per branch per plant per evaluated accession was recorded weekly during four flowering semesters. Subsequently, the phenological flowering descriptors, namely synchrony among individuals, intraindividual temporal variability and number of events were calculated. The data were analyzed descriptively, and then the inferential component was conducted using analysis of variance for a two-factor additive model and randomization restriction. The results showed that there are two flowering patterns according to the expression of flowering in the floral cycles, the “annual” class in the department of Cesar and the “continual” class in the departments of Caldas, Quindío and Cauca. The phenological descriptors show differences between the departments according to the coffee zone to which it belongs (northern, central or southern). In turn, the floral pattern of each area can be linked to the latitudinal change in daily sunshine, as well as to the distribution of rainfall and temperature, in a very broad sense and based on the literature. The data did not provide statistical evidence to suggest differences among the accessions or between the tree sizes evaluated.

## 1. Introduction

Of the 124 species that make up the genus *Coffea* [1], the species *Coffea arabica* L., a tetraploid, autogamous species adapted to altitudes above 1000 m, and *Coffea canephora* Pierre ex Froehner, a diploid, allogeneic species adapted to heights below 1000 m [2], are responsible for 58% and 42% of world production, respectively, with Brazil (36%), Vietnam (18%), Colombia (8%) and Indonesia (6%) being the main global producers of beans [3]. The traditional varieties of *Coffea arabica* L. most recognized in America include the tall varieties Tipica, Bourbon and Mundo Novo (long size—LS), reaching a height of 4–6 m in a growing cycle (five to six years), as well as the short varieties Caturra, Catuaí and some Catimore varieties (short size—SS) that reach a height of 2–3 m [4]. In Colombia, the coffee park is located between 0°55′22″ and 11°16′37″ N latitude and encompassed 842,420 hectares as of 2022, of which 86% are planted with improved varieties resistant to coffee rust (*Hemileia vastatrix* Berk. & Broome) [5]. Since 1982, with the launch of the Colombia variety, Cenicafé opted for the genetic diversity strategy that is based on the use of compound coffee varieties (Colombia, Tabi, Castillo^®^ General, Castillo^®^ Regionales, Cenicafé 1 and Castillo^®^ Zonales; Cenicafé, Manizales, Colombia). In these varieties, different lines with comparable phenotypic and production characteristics, but different resistance mechanisms against coffee rust are mixed [6].

Phenology is the study of the time of recurring natural phenomena in relation to climate [7]. This is affected both by abiotic factors such as photoperiodicity, precipitation, temperature and humidity, as well as by biotic factors such as genetics and intraspecific and interspecific competition for resources [8]. Different phenological phases affect coffee production, one of the most important of which is flowering [9]. Flowering begins with the swelling of the foliar axils in plagiotropic branches (stage BBCH 51), from which three to five differentiated buds emerge in different degrees of development and are covered by brown mucilage (stage BBCH 53). Then, four floral meristems asynchronous in their development form in each bud, giving rise to simple green flowers that are still closed and attached by their corollas (state BBCH 57) until they reach a size of 4 to 6 mm, at which point they separate and cease growing to enter a dormancy period (state BBCH 58). The dormancy period ends with the first rains that restart growth until the flowers reach 6–10 mm in length when they change from green to white but are still closed, a period known as preanthesis (state BBCH 59). This period ends three to four days later with the opening of the corolla and the flower reaching the anthesis period (state BBCH 59), in which the flower buds open, exposing the pistil and anthers [4,10]. Anthesis occurs early in the morning [11], and the flowers remain open on average for three days [12]. The floral structures are mainly concentrated in the upper middle third of the plant but change position with age since they move toward the apex as new plagiotropic branches grow.

The main causes of coffee flowering events and the resulting patterns in different regions of Colombia are asynchrony in the development of floral meristems [13], lack of well-defined dry seasons, and inductive photoperiod throughout the year in equatorial regions [10,14]. Arcila et al. [12] and Rendón et al. [15] have descriptively identified five floral patterns according to the distribution of the number of flowers in the two flowering periods (FP) [FP I: November–April and FP II: May–October] and the latitudinal location. Thus, in latitudes between 1 and 4° N, approximately 10% of total flowering occurs in FP I, while the remaining 90% occurs in FP II. For latitudes between 4 and 5° N, the flowering is distributed as 60% in FP I and 40% in FP II. In latitudes between 5 and 8° N, 75% of the flowers appear during FP I and the remaining 25% in FP II. In latitudes between 8 and 9° N, 90% of the flowers are produced during FP I, and only 10% during FP II, while in latitudes above 10° North, blooms only occur during FP I. In summary, above 4° N latitude, the heaviest blooms tend to occur during FP I, while below this latitude they occur during FP II.

Peña et al. [16] reported that over 10° north latitude, the preanthesis (stage BBCH 59) of the heaviest blooms (February and March) in FP I had an inverse association with the photoperiod recorded two or three months before the event, a time that corresponds to the month of December when the maximum decrease in day length occurs (11.5 h). This indicates that photoperiod affects the bud development process that starts with flower initiation until dormancy (stage BBCH 58). However, this process begins earlier, between October and November, shortly after the autumn equinox (the last week of September in Colombia), when the day length begins to decrease to its lowest point in December [17]. Under controlled conditions, it was possible to stimulate floral differentiation in coffee by experimentally decreasing the day length [18]. The induction of flower buds in trees of 25 species has been shown to be associated with a decrease in the photoperiod of 30 min or less [19]. Experiments with plants grown under artificial light and dark conditions have shown that the night length is the most important factor in inducing flowering, which is why “short-day plants” should be more appropriately referred to as “long night-plants” [20]. In contrast, the absence of blooms during FP II is explained by the inhibition of bud initiation with the increase in the day length (maximum 12.8 h) that occurs during this period. Later, once the flower buds reach dormancy, they require a dry period and its subsequent interruption by rains to complete their transition until preanthesis and anthesis, an event that occurs between the months of January and February at this latitude (10° N) [16]. Recent phenological studies in multiple tree species have shown that seasonal changes in daily insolation at the top of the atmosphere (hereinafter daily insolation) are the most consistent environmental signal inducing dormancy in flower buds in temperate, tropical and equatorial latitudes, surpassing day length [21]. Daily insolation is a function of the day length and radiation intensity integrated over a day without atmospheric effects, which vary throughout the year and with latitude, according to the angle with which the sun’s rays hit the Earth’s surface [22]. The changes in day length or daily insolation trigger the molecular processes that lead to possible subsequent changes in the development of plant organs, making the variation in light intensity due to cloudiness irrelevant. In contrast, the absorption of irradiation in photosynthesis provides energy gains, which are strongly affected by clouds [23].

However, between 4.5° and 7.0° N latitude, where the decrease in day length during FP I (minimum 11.8 h) and its increase during FP II (12.4 h) are slight and even insignificant below 4.5° N for FP I (minimum 12 h), the day length does not seem to be associated with the initiation of flower buds and the subsequent anthesis [14]. In this case, dry periods of moderate magnitude (approximately two weeks) and their subsequent interruption by rains in the months of January to February and June to August seem to be the factor most consistent with the largest blooms recorded from January to March and July to September [10]. However, flowering events are not restricted to only these months; they occur throughout the year given the adequate distribution of rainfall without long dry periods [24]. The stress required to break dormancy in flower buds and restart their growth is still controversial since anthesis in coffee can be induced by a short period with a severe water deficit (predawn leaf water potential, Ψpd: −2.6 MPa), as well as by a long period with moderate water deficit (Ψpd: −0.8 MPa) [25]. This results in uneven fruit maturation and consequently in the distribution of harvesting events per year [26]. Additionally, the Intertropical Convergence Zone (ITCZ) influences the total accumulated rainfall per period; thus, the regions above 4.5° N present less accumulated rainfall during FP I and a greater quantity of flowers, while in those below 4.5° N, this occurs during FP II [16,27].

In perennial crops, the evaluation of flowering phenology provides clues about the functional attributes of tree reproduction [28]. In the case of flowering, a descriptive way to characterize flowering patterns is through the classification proposed by Newstrom et al. [29], which is based on the frequency of the number of cycles with flowering events per year and considers four kinds of patterns: “continual” (flowering with brief sporadic pauses), “sub-annual” (blooms in more than one cycle per year), “annual” (only one main cycle per year) and “supra-annual” (one cycle for more than one year). Conversely, a quantitative characterization of flowering requires censusing open flowers and recording flowering events for the calculation of flowering descriptors such as the beginning of flowering, end of flowering, duration of flowering, number of flowering events, the amplitude of flowering, temporal variability and flowering synchrony, both at the population level and at the level of individuals within populations [8,29,30,31]. Of these descriptors, the intraindividual temporal variability is associated with the different amplitudes of flowering events during a season, using the coefficient of variation as a metric, while the synchrony between individuals measures the overlap of the temporal distribution of an individual’ flowering in relation to that of other individuals [32]. In this sense, intraindividual variability and synchrony among individuals have been used to evaluate flowering patterns from a viewpoint closer to their biological causes [33]. Therefore, they are more suitable when the reproductive structures are grouped temporarily or when the pattern does not show a unimodal distribution [32]. This is the case for coffee flowering, due to the different degrees of development of the floral meristems [13] and the different flowering distribution patterns of the different coffee zones of Colombia [15].

The phenological comparison between plants with different characteristics facilitates the analysis of the factors that influence the phenology of flowering the most [34], which can facilitate the planning and/or development of new strategies that allow greater profitability and sustainability in coffee cultivation [35]. Coffee tree phenology has been a useful tool to define the times of practices, such as fertilizer application, pest and disease control, and weed management, among other tasks [4]. Based on the above, this study aims to answer the following questions: (1) Do flowering patterns and their descriptors vary among different *C. arabica* L. accessions? (2) Do the flowering patterns and their descriptors change with geography in Colombia? It is the first paper where the synchrony and temporal variability of flower patterns of *C. Arabica* are quantitatively examined by genotype and location in Colombia.

## 2. Materials and Methods

### 2.1. Study Site and Plant Material

The study was carried out in four Experimental Stations of the National Coffee Research Center—Cenicafé, located in the departments of Cesar (Pueblo Bello municipality, 10°25′18″ N, 73°34′29″ W, 1134 m), Caldas (Chinchiná municipality, 4°58′19.1″ N, 75°39′8.2″ W, 1407 m), Quindío (Buenavista municipality, 4°23′44″ N, 75°44′3″ W, 1203 m) and Cauca (El Tambo municipality of, 2°24′17.4″ N, 76°44′30.1″ W, 1755 m) of Colombia (Figure 1). These departments represent the northern (Cesar), central (Caldas and Quindío) and southern (Cauca) coffee zones of the country (FNC, [36]). In each locality, 15 accessions of *C. arabica* L. from the Colombian Coffee Collection (CCC) were evaluated, of which four are long (LS) and 11 are short (SS). The accessions evaluated are genetically diverse according to the characterization of 13,204 SNPs distributed throughout the genome of *C. arabica* L. The LS accessions correspond to three materials of Ethiopian origin (E057 = CCC168, E338 = CCC354 and E554 = CCC534) and the Tabi variety. The SS accessions are advanced lines of the genetic improvement program identified as BH1247, CU1812, CU1983, CU1990, CU1991, CU1993, CX2197, CX2385, CX2720, CX2848 and CX2866. These represent the diversity of the improved lines that make up the Castillo^®^ General variety. The accessions were planted in May 2015 in two adjacent experimental lots according to their size, as follows: the LS group was planted at a distance of 2.0 m between rows and 1.5 m between plants, whereas the SS group was planted at 1.5 m between the rows and 1.0 m between the plants. In each experimental lot per department, 30 plants were planted per accession. In all localities, with the exception of Pueblo Bello, the plants were established under full sun exposure; however, in the departments of Quindío and Cauca, transitory tree cover was maintained during the first eight months of establishment in the field. In Pueblo Bello, the plants were grown under an agroforestry system with guamo (*Inga edulis* (Vert) Mall) established at a distance of 12 m × 12 m, generating 45% shade. In this context, the plants represent the individual level, and all the plants in the accession as a whole represent the population level.

### 2.2. Data Recording

In each of the plants per accession evaluated, four plagiotropic branches of the upper middle third (productive third) were selected and marked one week before the start of the FP, representing the four cardinal points. The selected branches had flower buds in a dormant state (BBCH 58). The number of flower buds in the preanthesis stage (BBCH 59) was recorded per individual and week (x_ti_) during the entire floral semester, following the methodology established by Rendón et al. [37]. The representativeness of the branches with respect to the flowering of the entire plant was assumed based on previous studies [38]. The counts were not performed in the anthesis state because the open corollas make it difficult to count the flower buds, especially in very abundant flowering events where sampling errors can be incurred. The flower buds were recorded during four flowering semesters: (1st) November 2016–April 2017; (2nd) May 2017–October 2017; (3rd) November 2017–April 2018; and (4th) May 2018–October 2018, which are framed in FP I (November to April) and FP II (May–October). In the records, the number of individuals (plants) was different depending on the department and the flowering semester because some trees died before the data were collected and also after; in other cases, there were individuals that did not flower during the semester, so they were excluded. In this way, the number of individuals analyzed varied between 10 and 20 depending on the accession and the semester in the departments of Cesar, Caldas and Cauca, whereas in the department of Quindío, where mortality was high in the establishment of the crop, the number of individuals ranged from five to ten. The monitoring of the daily meteorological conditions of each site was carried out using the remote automatic weather station RAWS-F (FireWeather, Campbell Scientific Inc., Logan, UT, USA) of the Coffee Meteorological Network. The meteorological information of the climatic stations is listed in Table 1. The day length and daily insolation data were obtained from NASA’s E AR5 simulation model, and both variables are available on its website (https://data.giss.nasa.gov/modelE/ar5plots/srlocat.html, accessed on 26 January 2022) (Table 1).

### 2.3. Phenological Flowering Descriptors

The descriptors were calculated from the weekly records of the number of flower buds at the individual or population level for each accession by floral semester, depending on the case. The flowering synchrony among individuals (r_i_) was estimated as the average for the absolute values of all the pairwise Spearman correlations of the number of flower buds per week (x_ti_) of each of the individuals within the population [30]. The intraindividual temporal variability of flowering (CV_i_) was calculated as the average of the individual coefficients of variation in x_ti_ over the n weeks of the flowering semester for each of the individuals within the population [30]. Weeks without flower buds were treated as zeros. Finally, the number of flowering events at the population level (Event) was estimated as the number of weeks of the flowering semester when the accession presented at least one flower bud, regardless of the individual.

### 2.4. Data Analysis

The flowering patterns by accession, floral semester and department were described using bubble diagrams in the floral calendars, which showed the number of flower buds in each week of the semester per accession, in addition to the total number of flower buds per semester. The semester data of the descriptors CV_i_ and r_i_ were discriminated by accession and department. For the inferential component, a linear model with three main effects (Site, Semester and Accession) and their interactions was proposed using the sum of squares of Type II in the analysis of variance (AOV) [39]. However, this first model showed that the main effect of Genotype and the associated interactions (Site × Accession and Semester × Accession) did not present statistical evidence against the null hypothesis of null effect. Therefore, a second linear model with two main effects (Site, Semester) and their interaction was developed for the descriptors r_i_ and CV_i_ using the same type of sum of squares for the AOV. The interaction (Site × Semester) was broken down first by analyzing the simple effect of the adjusted semester by the Bonferroni method and then by analyzing all the multiple comparisons by pairs between the different sites organized by semester using the t test adjusted via the Bonferroni method [40]. To reduce the risk of false positives by having several separately modeled responses, the threshold for significant differences in all procedures was *p* = 0.005 [41,42,43]. The assumptions of normality (Shapiro—Wilk test) and homogeneity of variances (Levene’s test) in each AOV were met (*p* > 0.05). In the case of the event descriptor, the analysis was carried out descriptively using the median of the data. The “car” [44,45], “ggplot2” [46], “psych” [47], “rstatix” [48] and “stats” packages from the R software [49] version 4.2.3 were used for data analysis.

## 3. Results and Discussion

### 3.1. Flowering Patterns in Coffee

The floral calendars of *C. arabica* show that the flowering semesters by the department were expressed differently (Figure 2). In the departments of the central (Caldas and Quindío) and southern (Cauca) coffee zones, the blooms occurred both in the FP I (November–April) and FP II (May–October) flowering semesters (Figure 2e–l,n–p), whereas in the northern coffee zone (Cesar), the blooms only occurred in FP I (1st and 3rd semesters) (Figure 2a,c). It is worth mentioning that the absence of records during the first semester of flowering in Cauca is due to the lower development of the crop (still in the vegetative stage), given the lower temperatures in this locality compared to the others, and not to a differential expression of flowering, as occurs in Cesar, where flowering is only expressed in FP I (Table 1). The above is supported by the historical records of coffee flowering in Colombia [15,16]. The presence or absence of flower buds per period had an impact on the kinds of flower patterns reported for the accessions by department, according to the Newstrom classification [29]. In the department of Cesar (the northern coffee zone), the flowering pattern observed was the “annual” type [29], characterized by discontinuous and dispersed blooms in very few flowering events per FP I (two events) within a short period of time (one to two months) with one cycle per year (Figure 2a–d). In the departments of Caldas, Quindío (the central coffee zone) and Cauca (the southern coffee zone), the “continual” type pattern was observed [29], with many short-lived flowering events scattered throughout the year that only cease sporadically and briefly (with blooms in both FP I and FP II) (Figure 2e–l,n–p). However, it should be noted that in the department of Cauca, the number of flowering events was lower than that observed in Caldas and Quindío (between 11 and 15 fewer events). The LS and SS accessions in each department did not show relevant differences in their flowering patterns (i.e., an accession with an “annual” pattern in Cauca or an accession with seven or eight flowering events in Caldas). In the departments of Caldas and Quindío, the total number of flower buds per semester tended to be higher during FP I and lower during FP II (Figure 2e–l), whereas in Cauca, the opposite occurred (Figure 2n–p).

The comparison of the flowering patterns among the departments that make up the northern (Cesar), central (Caldas and Quindío) and southern (Cauca) coffee zones showed that the differences among them are largely related to latitudinal change. Variations in coffee flowering patterns due to latitudinal location have been previously reported in Colombia [12,15]. In Cesar (10° N), the short-lived blooms in FP I and none in FP II show an “annual” type pattern, according to the classification of Newstrom et al. [29]. In the rest of the departments evaluated, blooms were observed in both FP I and FP II in practically all months of the year, which is why their floral pattern was classified as a “continual” type (Figure 2e–l,n–p). The number of total flowers in Figure 2 was inversely associated with the accumulated rainfall and the number of days with rain per period; thus, in the departments of Cesar and Caldas, where the highest number of flowers was observed in FP I (Figure 2a,c,e,g), rainfall was lower (Table 1), while the opposite occurred in Cauca (Figure 2n,p;Table 1). This is mainly due to the movement of the ITCZ, which determines the occurrence of rains and their magnitude during the year [27]. Phenological observations of *C. arabica* (Tico and Caturra hybrid cultivars) in Ciudad Colón (Costa Rica) at 9°54′ and 84°14′ W showed that blooms only occur between the months of December and April [50], as in Lagunas (Philippines), located at 14°09′ N and 121° 15′ W [35]. In Ruiru (1° S; Kenya), the coffee has two flowering periods per year [51], as occurs in Colombia at latitudes lower than 10° N. At 10° N, the short-day species *Cordia alliodora* and *Gliricidia sepium* bloom once a year at the end of December, but near the equator at 3° N, they bloom twice a year, in May and in November [23].

### 3.2. Effect of Accessions and Geographic Location on Phenological Descriptors of Flowering

The AOV of the linear model with three main effects for the descriptors r_i_ and CV_i_ indicated that there is no statistical evidence against the null hypothesis of null effect for the accession source of variation or in the interactions involved with it [r_i_: Genotype F = 2.0575, *p* = 0.02256; Site × Genotype F = 0.8086, *p* = 0.77403; Semester × Genotype F = 0.6751, *p* = 0.91912. CV_i_: Genotype F = 2.16, *p* = 0.016, Site × Genotype F = 0.73, *p* = 0.867, Semester × Genotype F = 0.77, *p* = 0.825]. The general differences between the accessions for the descriptor r_i_ were no greater than nine percentage points between CX2197 (r_i_ = 0.72) and the variety Tabi (r_i_ = 0.81) (Figure 3a), while for the descriptor CV_i_, this difference did not exceed 11% between E338 (CV_i_ = 2.89) and CU1983 (CV_i_ = 2.59) (Figure 3b). The AOV was performed separately because the r_i_ and CV_i_ descriptors are linked to two different biological concepts [32], which are necessary for the interpretation of flowering patterns (variability within plants and synchrony among plants). The few variations between them in the event descriptor were not greater than one event with respect to the general median (15 events) (Figure 3c). This indicates that the differences in flowering patterns are not linked to the accessions evaluated, although they have genetic diversity. In the same way, it also allows us to deduce that the size of the plants (LS and SS) had no effect on the flowering phenology from the statistical point of view (Figure 3). Previous studies reported that the flowering initiation patterns in nine coffee cultivars (Red Catuai, Catuai, Yellow Caturra, Red Caturra, Catimor, BMK, SL6, K7 and LB) did not show significant variations, which affects synchronized flowering [52]. However, reports of contrasting flowering patterns at the same latitude exist in *C. arabica* L., such as the *Semperforens* mutation, which has continuous flowering events throughout the year in Brazil [53], when the common pattern for commercial varieties such as Bourbon is to bloom two to three times per year [54], but it is the only special case reported.

The AOV of the linear model with two main effects for the descriptors r_i_ [Site: F = 115.141, *p* = 7.82 × 10^−42^; Semester: F = 28.359, *p* = 4.30 × 10^−15^] and CV_i_ [Site: F = 829, *p* = 7.9 × 10^−106^; Semester: F = 48, *p* = 5.6 × 10^−23^] showed that there is statistical evidence against the null hypothesis for the Site x Semester interaction [r_i_: F = 20.129, *p* = 5.40 × 10^−18^; CV_i_: F = 33.0, *p* = 6.5 × 10^−27^]. Consequently, for the two descriptors mentioned, the Site x Semester interaction was decomposed. First, the analysis of the simple effects for Semester showed that there were significant differences in the mean of the sites for each of the evaluated semesters, both for r_i_ [1st: F = 59.3, *p* = 2.33 × 10^−12^; 2nd: F = 58.2, *p* = 3.11 × 10^−12^; 3rd: F = 45.2, *p* = 2.25 × 10^−14^; 4th: F = 48.5, *p* = 4.92 × 10^−11^] as for CV_i_ [1st: F = 343.0, *p* = 3.81 × 10^−26^; 2nd: F = 22.5, *p* = 9.16 × 10^−7^; 3rd: F = 514.0, *p* = 4.12 × 10^−40^; 4th: F = 343.0, *p* = 3.78 × 10^−26^].

Then, for the descriptor r_i_, the analysis of all the pairwise comparisons between the different sites organized by semester showed that the department of Cesar achieved the highest flowering synchrony with respect to the other departments in statistical terms in the 1st and 3rd semester (between 12 and 24 percentage points) (Figure 4). Cauca, with the second highest synchrony, was statistically higher than Caldas in the 2nd and 4th semesters (between 25 and 27 percentage points), whereas with respect to Quindío, it was only higher in the 2nd semester (26 percentage points). Finally, none of these three departments showed statistical differences in the 3rd semester (Figure 4). The synchrony in Caldas was statistically greater than that in Quindío during the 1st semester (9 percentage points), but this was reversed in the 4th semester when Quindío had greater synchrony (19 percentage points) for the remaining semesters (2nd and 3rd). Between Caldas and Quindío, no differences were observed from the null hypothesis (Figure 4).

Similarly, the pairwise comparisons for the CV_i_ descriptor showed that the department of Cesar statistically obtained the highest temporal variability in the flowering patterns of the 1st and 3rd semesters (between 43 and 65%) (Figure 5), followed in order by the temporal variability achieved in Cauca, which was statistically greater than that found in Caldas during the 2nd, 3rd and 4th semesters (between 20 and 39%), whereas in Quindío, it was observed in the 2nd and 4th semesters (20 and 44%, respectively) (Figure 5). During the 3rd Semester, the department of Quindío presented a statistically higher CV_i_ than that of Caldas [31%]; however, in the 1st, 2nd, and 4th semesters, no difference was observed in the temporal variability of these departments (Figure 5).

The event descriptor descriptively showed an inverse behavior to that observed in the r_i_ and CV_i_ descriptors by showing that the departments of Caldas and Quindío presented between 2.2 and 4.0 times the number of flowering events than Cauca, and between 7.5 and 13.0 times the number recorded in Cesar. Likewise, the department of Cauca had 5.0 times the number of flowering events than those found in Cesar (Figure 6).

The “annual” type flowering pattern in the northern coffee zone (Cesar) was characterized by a very heterogeneous distribution (very high or low peaks) with very few flowering events during a short time interval during FP I (Figure 2a,c and Figure 6). Hence, the intraindividual temporal variability for this department is very high, and since the possibility of flowering of a plant in conjunction with others increases with fewer flowering events, the flowering synchrony among individuals was very high (Figure 4 and Figure 5). The occurrence of blooms only during FP I (Figure 2a,c) has been commonly linked to the considerable reduction in day length (more than 30 min) that begins at the autumn equinox (September) and extends until the winter solstice (December) [16] and with the first rains after a defined dry period that cause the breaking of the dormancy of the flower buds in coffee [55], which occurs between December and March for this department [27]. Conversely, the absence of blooms during FP II between May and October in Cesar (Figure 2b,d) has been associated with the increase in the number of light hours toward the summer solstice (June), although more recent phenological studies point to the daily insolation plateau that forms during this period [16,21,23]. In tropical areas far from the equator (>10° N or >10° S), coffee exhibits one to three flowering events per year [14,24]. Other species in which the “annual” type pattern has been reported are *Myrcia rhodosepala* and *Blepharocalyx salicifolius* [56] of the family Myrtaceae and in the species *Juncus effusus* and *Juncus tenuis* of the family Juncaceae [32].

The “continual” flowering patterns in the departments of central (Caldas and Quindío) and southern (Cauca) coffee zones cover both FP I and FP II (Figure 2e–l,n–p). This expression of flowering in both FP has been attributed by other phenological studies to the double decrease in daily insolation during the year, which occurs at latitudes below 10° N and above 10° S [21,23]. In this sense, the decrease in insolation acts by “activating” flowering expression in this period, whereas a plateau in insolation “deactivates” flowering expression, as occurs in Cesar (Figure 2). In conspecific trees growing in tropical and equatorial latitudes, the induction of dormancy in flower buds and flowering has been shown to be more closely related to seasonal changes in the variation of daily insolation than to changes in the day length [21]. In the departments of Caldas and Quindío, the blooms occurred during all the months of the year, causing their distribution to be more homogeneous during the FP in the multiple flowering events that it exhibits (Figure 2 and Figure 6), which leads to low or moderate intraindividual temporal variability (the magnitudes of the peaks are less contrasting). In turn, with the increase in flowering events of low magnitude per FP, the possibility of flowering of a plant in conjunction with others decreases, making the flowering synchrony among individuals lower (Figure 4 and Figure 5). In equatorial regions, coffee flowering events occur at any time of the year at intervals of two to several weeks due to the lack of well-defined dry and wet periods [38,55], as occurs in the central coffee zone, where the FP shows a similarity of days with rain between periods (Table 1), which enables a high number of flowering events in coffee plantations [24]. Rains during the late stages of bud development affect the flowering process of coffee, causing an increase in the number of flowering events, or on the contrary, can reduce them when the frequency of rains is lower and there are dry periods between rains [14,24]. The “continual” flowering pattern has also been observed in the flowering of *Guatteria aeroginosa* of the family Annonaceae [29] and in the cactus *Neolloydia conoidea* of the family Cactaceae [57].

In the department of Cauca, located in the southern coffee zone, the blooms present a “continual” pattern (Figure 2n–p) as in the departments of Caldas and Quindío (Figure 2e–l), but unlike these, in Cauca, the flowering synchrony among individuals and the intraindividual temporal variability tended to be greater (Figure 4 and Figure 5). This is because the distribution of flowering in Cauca (Figure 2n–p) is more heterogeneous than that observed in Caldas and Quindío (Figure 2e–p) by having fewer flowering events (Figure 6), which increases the flowering synchrony among individuals and intraindividual temporal variability (Figure 4 and Figure 5). The lower number of flowering events in Cauca could be linked to the lower temperatures in this area with respect to Caldas and Quindío (Figure 6; Table 1). Cold periods have an impact on more synchronous blooms by increasing the number of flower buds that reach dormancy but delaying later stages such as anthesis, which occurs in greater proportion with rehydration [14]. This, in turn, would help reduce flowering expression at other events and therefore reduce the number of flowering events. In *Spartina alterniflora*, changes in the mean temperature due to latitudinal variation have been inversely associated with flowering synchrony since the increase in flowering synchrony can help individuals to avoid the damage caused by low temperatures to the reproductive organs [58]. Temperatures of 23 °C during the day and 18 °C at night favor the synchronous development of flower buds in coffee (a range close to that found in Cauca, Table 1) and maximize the number of inflorescences per node with respect to higher temperatures (28 °C during the day/23 °C at night) [52].

The flowering patterns of the departments and their phenological descriptors vary greatly by coffee zone (northern, central, and southern) where they are located, which is linked to latitude, and the characteristics of each semester, which are associated with environmental variation. Daily insolation, rainfall distribution and temperature changes can be linked to the phenological descriptors of the patterns by department in a very broad sense based on the literature. However, the expression of flowering in the field is the result of the interaction of these and other environmental factors in each zone and time, which exceeds the scope of this study and should be considered in future research. The recognition of flowering patterns and their quantification through phenological descriptors has important agronomic implications for the harvest since the flowering events will produce subsequent peaks of fruit collection, as long as the fruit development time is constant [59], in addition to allowing future comparative studies of flowering patterns in quantitative terms.

## 4. Conclusions

The flowering calendars of *C. arabica* L. in Colombia showed two classes of patterns according to the FP regardless of the size of the plant or the accession, the “annual” class in the department of Cesar (northern coffee zone) with one cycle per year, and the “continual” class in the departments of Caldas, Quindío (central coffee zone) and Cauca (southern coffee zone) with brief interruptions in the flowering. The statistical analysis of the phenological descriptors r_i_ and CV_i_ and the descriptive descriptor event allowed us to quantitatively characterize the differences in the flowering patterns for each department and associate them with their corresponding coffee zone. In turn, the flowering pattern of each area can be linked to the latitudinal change in daily sunshine and the distribution of rainfall, in addition to temperature. The differences in flowering expression result mainly from the environmental component (flowering semester and location), but not the accessions or their different sizes. The results of the present study contribute to the previously acquired knowledge about the flowering patterns of *C. arabica* L. in Colombia by quantitatively establishing the changes that they present through the different coffee growing areas and providing a comparative basis for future phenological studies.

## Figures and Tables

**Figure 1 plants-12-03332-f001:**
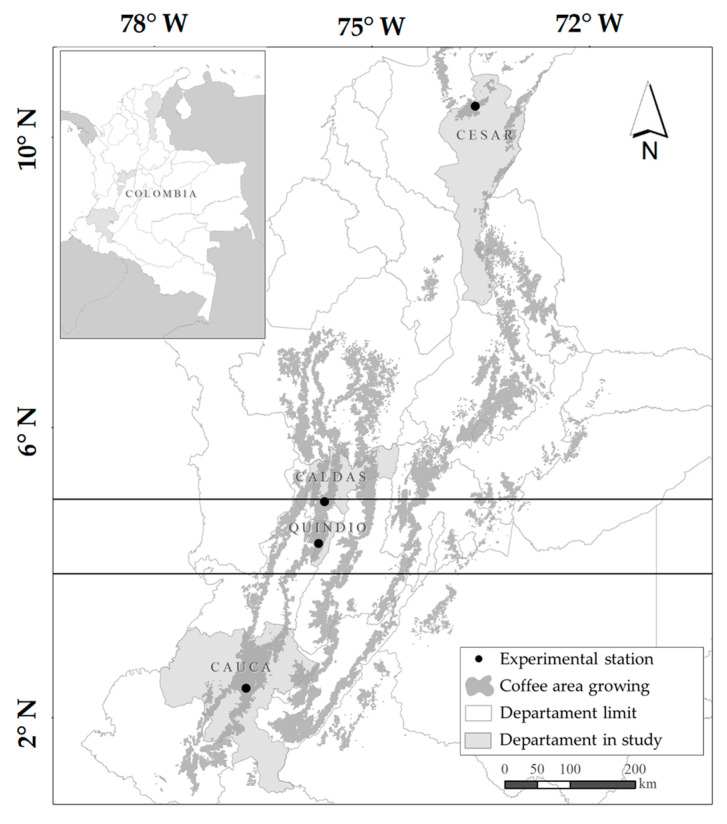
Colombian coffee zone and location of the Pueblo Bello (Cesar department), Naranjal (Caldas department), Paraguaicito (Quindío department) and El Tambo (Cauca department) experimental stations where the experimental lots were established.

**Figure 2 plants-12-03332-f002:**
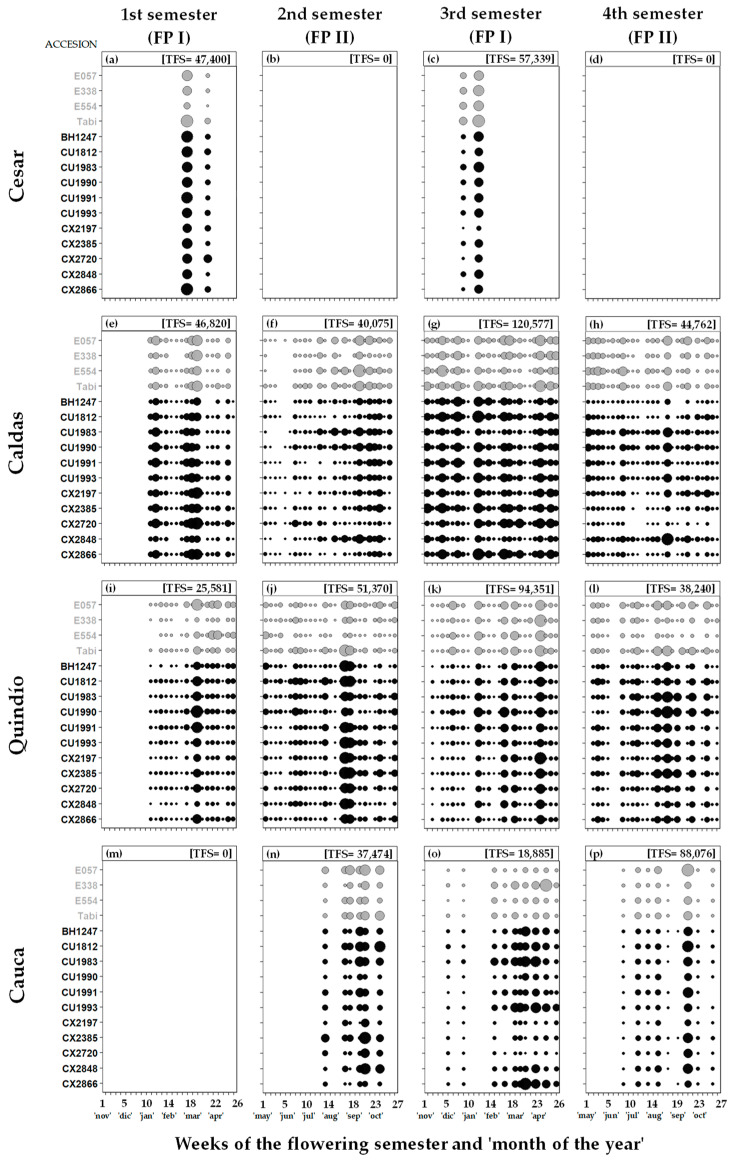
Floral calendars for 15 accessions of *C. arabica* L. located in four departments of Colombia during four flowering semesters: 1st Semester [November 2016–April 2017] (**a**,**e**,**i**,**m**), 2nd Semester [May 2017–October 2017] (**b**,**f**,**j**,**n**), 3rd Semester [November 2017–April 2018] (**c**,**g**,**k**,**o**), 4th Semester [May 2018–October 2018] (**d**,**h**,**l**,**p**). The 1st and 3rd semesters correspond to the first flowering period (FP I), whereas the 2nd and 4th semesters correspond to the second flowering period (FP II). The accessions are classified into long-size (LS; accession letters and bubbles in gray) and short-size (SS; accession letters and bubbles in black) plants. The bubbles indicate the weeks of the flowering semester with observation of flower structures (buttons in preanthesis), and the size of the bubbles is proportional to the number of flower buds recorded in each accession by sampling date. The total number of flower buds per semester (TFS) is shown in the upper box of each figure.

**Figure 3 plants-12-03332-f003:**
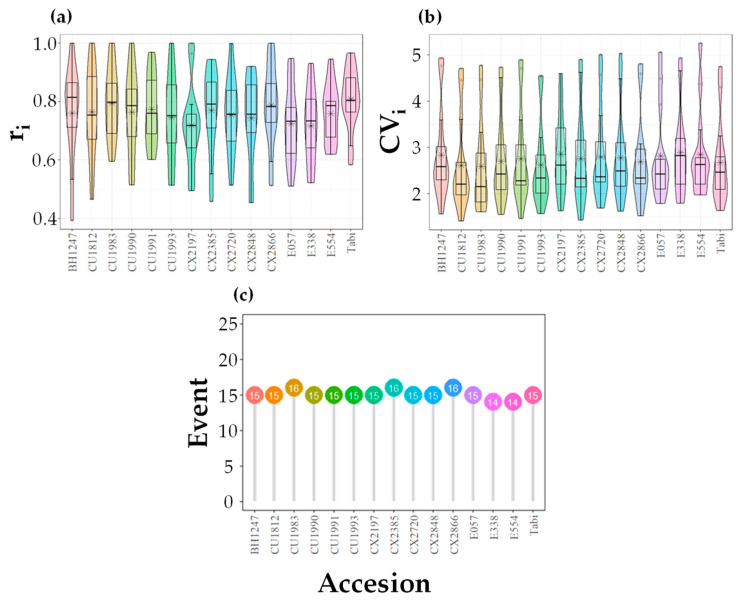
Distribution of the descriptors flowering synchrony among individuals [r_i_] (**a**), intraindividual temporal variability [CV_i_] (**b**), and median number of flowering events [Event] (**c**) per accession in *C. arabica* L. plants. The asterisk symbol corresponds to the mean.

**Figure 4 plants-12-03332-f004:**
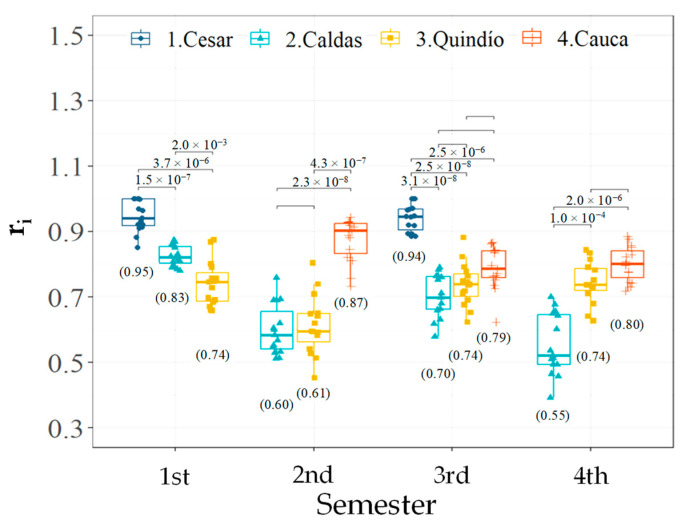
Multiple pairwise comparisons (⌐¬) between departments per semester for the descriptor flowering synchrony among individuals [r_i_] in *C. arabica* L. plants. Comparisons with values in scientific annotation indicate statistical differences in favor of the alternative hypothesis. The graphs show the distribution of the data, while the value in parentheses corresponds to the mean.

**Figure 5 plants-12-03332-f005:**
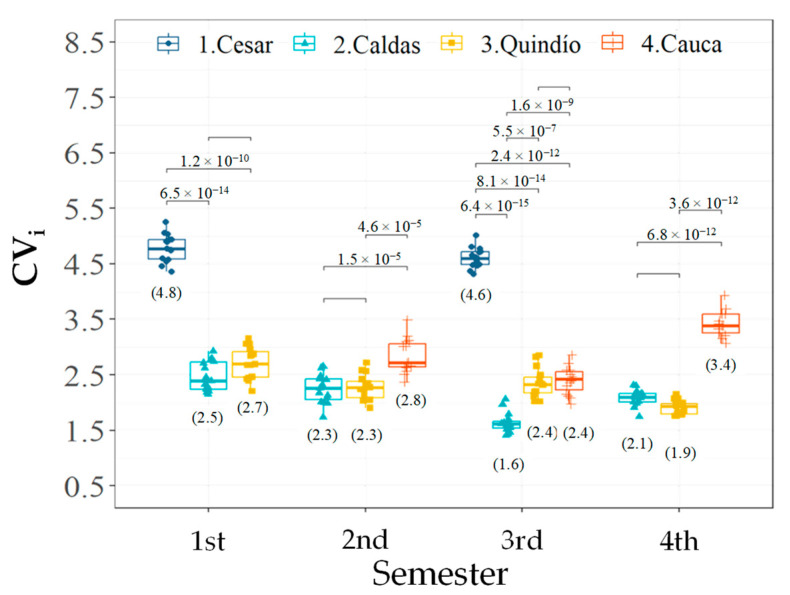
Multiple pairwise comparisons (⌐¬) between departments per semester for the descriptor intraindividual temporal variability [CV_i_] in *C. arabica* L. plants. Comparisons with values in scientific annotation indicate statistical differences in favor of the alternative hypothesis. The graphs show the distribution of the data, while the value in parentheses corresponds to the mean.

**Figure 6 plants-12-03332-f006:**
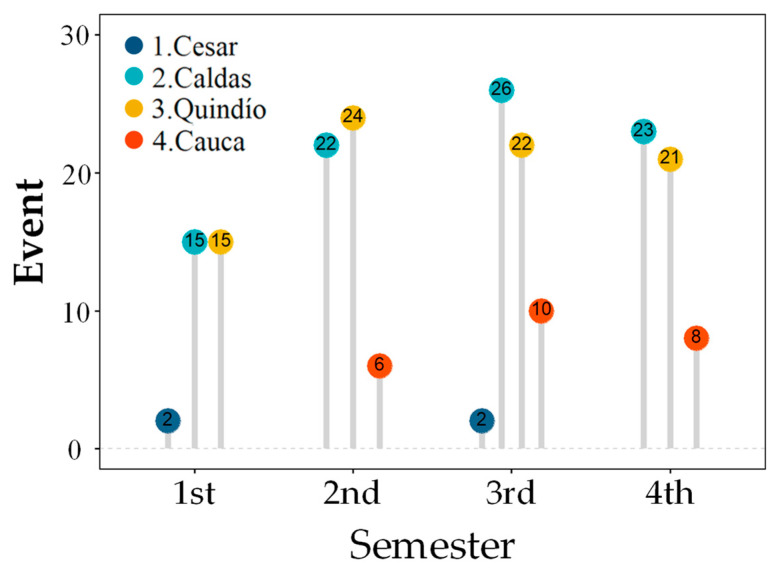
Distribution of the number of flowering events [Event] by department for each semester recorded in *C. arabica* L. plants. The values correspond to the median of the data.

**Table 1 plants-12-03332-t001:** Mean air temperature (Temp_Mean_, °C), maximum air temperature (Temp_Max_, °C), minimum air temperature (Temp_Min_, °C), accumulated daily insolation (Insolation, W m^−2^), number of accumulated light hours (Daylength, W m^−2^), accumulated rainfall (Rainfall, mm) and number of days with rain (DaysRain, #) per flowering semester (Sem) for the four departments (DEP) evaluated in Colombia.

DEP	Sem	Temp_Mean_	Temp_Max_	Temp_Min_	Insolation	Daylength	Rainfall	DaysRain
Cesar	1st	20.4	27.7	14.9	71,475	2142	680	58
	2nd	21.8	28.0	17.2	78,602	2285	1852	100
	3rd	22.9	27.5	17.2	71,463	2142	656	46
	4th	21.3	28.0	16.5	78,614	2285	1171	77
Mean		21.6	27.8	16.5	75,039	2214	1090	70
Caldas	1st	20.9	28.1	16.7	74,647	2169	1501	101
	2nd	21.2	28.6	17.0	77,194	2256	1793	118
	3rd	20.9	28.1	16.9	74,643	2169	1514	104
	4th	21.2	28.6	16.9	77,198	2256	1273	107
Mean		21.1	28.4	16.9	75,921	2213	1520	108
Quindío	1st	21.7	29.1	17.2	74,947	2172	1359	88
	2nd	22.2	29.8	17.6	77,008	2253	952	73
	3rd	21.7	29.5	17.4	74,944	2172	1707	92
	4th	22.0	29.7	17.5	77,011	2253	1040	75
Mean		21.9	29.5	17.4	75,978	2213	1265	82
Cauca	1st	19.3	24.9	15.4	75,930	2182	1420	107
	2nd	19.0	26.8	13.7	76,309	2242	748	65
	3rd	18.9	26.3	15.1	75,930	2182	1529	110
	4th	19.1	27.1	13.8	76,309	2242	601	71
Mean		19.1	26.3	14.5	76,120	2212	1075	88

## Data Availability

Not applicable.

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
