# Peer review of "Quantification and Qualification of Floral Patterns of *Coffea arabica* L. in Colombia"

_plants, 2023, doi:10.3390/plants12183332_

Round 1
Reviewer 1 Report
Dear Authors
I read the article Quantification and qualification of floral patterns of Coffea arabica L. in Colombia.
please revise the introduction part ( do not need figures)
please add novelty of your research in introduction part.
please add version R for software in Materials and method.
good luck
Author Response
Please revise the introduction part (do not need figures)
R/.
The figures were removed from the document and their corresponding description of the Introduction. The length of the Introduction was reduced from 2360 words to 1801 words (23% less).
Please add the novelty of your research in the introduction part.
R/.
A brief paragraph describing the novelty of the research was added in lines 171-173.
Please add version R for software in Materials and Method.
R/.
The software version was added on line 276.
Reviewer 2 Report
Very long introduction and parts of the text written as if they were results and discussion. It seems inappropriate.
Very long conclusion. The author could go straight to the point.
Most of the bibliographical references have more than 10 years of publication and are in exaggerated number. Maybe it can be reduced.
Author Response
Very long introduction and parts of the text written as if they were results and discussion. It seems inappropriate.
R/.
The length of the introduction was reduced from 2360 words to 1801 words (23% less), and the section written as results and discussion was withdrawn from the document. The figures were removed from the paper and their corresponding description.
Very long conclusion. The author could go straight to the point.
R/.
The length of the Conclusion was reduced from 232 words to 199 words (14% less), with some modifications in the text.
Most of the bibliographical references have more than 10 years of publication and are in exaggerated numbers. Maybe it can be reduced.
R/.
The bibliographic references were modified throughout the document. The changes are summarized in the following table:
|
|
References in the previous version |
|
References in the actual version |
||
|
Year |
Number |
% |
|
Number |
% |
|
2023-2014 |
24 |
34 |
|
30 |
51 |
|
2013-2004 |
19 |
27 |
|
14 |
24 |
|
>2003 |
27 |
39 |
|
15 |
25 |
Reviewer 3 Report
I have no further comments, but, looking at the results obtained, it would be very interesting to evaluate whether, depending on the frequency of flowering throughout the year in the different populations, it could have an effect on the yield or quality of the fruits obtained, thinking perhaps that that the more reproductive events the populations have, there could be some negative effect on the quality of the fruits. This for future studies.
Author Response
R/.
None.
Round 2
Reviewer 2 Report
We consider that the changes made are satisfactory.